# Synthesis and Characterization of Cu_2_ZnSnSe_4_ by Non-Vacuum Method for Photovoltaic Applications

**DOI:** 10.3390/nano12091503

**Published:** 2022-04-28

**Authors:** Meenakshi Sahu, Vasudeva Reddy Minnam Reddy, Bharati Patro, Chinho Park, Woo Kyoung Kim, Pratibha Sharma

**Affiliations:** 1Department of Energy Science and Engineering, Indian Institute of Technology Bombay Powai, Mumbai 400076, India; meenakshisahu.chem@gmail.com; 2Korea Institute of Energy Technology (KENTECH), 200 Hyukshin-ro, Naju 58330, Korea; 3School of Chemical Engineering, Yeungnam University, Gyeongsan 38541, Korea; drmvasudr9@gmail.com; 4Centre for Research in Nanotechnology and Sciences, Indian Institute of Technology Bombay Powai, Mumbai 400076, India; bharati@iitb.ac.in

**Keywords:** Cu_2_ZnSnSe_4_, thin film, selenization, photovoltaic devices applications

## Abstract

Wet ball milling was used for the synthesis of Cu_2_ZnSnSe_4_ (CZTSe) nanoparticles with a kesterite structure. The prepared nanoparticles were used for ink formulation. Surfactants and binders were added to improve the ink stability, prevent agglomeration, and enhance ink adhesion. The films deposited via spin coating were annealed at different temperatures using a rapid thermal processing system in the presence of selenium powder in an inert environment. Analytical techniques, such as X-ray diffraction, Raman spectroscopy, and Fourier-transform infrared spectroscopy, were used to confirm the formation of CZTSe nanoparticles with a single-phase, crystalline kesterite structure. Field-emission scanning electron microscopy and energy-dispersive X-ray spectroscopy were used to study the surface morphology and chemical composition of the thin films before and after annealing, with and without the sodium solution. The optoelectrical properties were investigated using ultraviolet-visible spectroscopy and Hall measurements. All the prepared CZTSe thin films exhibited a p-type nature with an optical bandgap in the range of 0.82–1.02 eV. The open-circuit voltage and fill factor of the CZTSe-based devices increased from 266 to 335 mV and from 37.79% to 44.19%, respectively, indicating a decrease in the number of recombination centers after Na incorporation.

## 1. Introduction

The quaternary inorganic semiconductor Cu_2_ZnSnSe_4_ (CZTSe) has unique optical and electrical properties. Consequently, they have emerged as ideal solar cell absorbers, replacing Cu(In, Ga)Se_2_ (CIGS) and CdTe-based thin-film photovoltaic devices [1,2]. Thus far, CZTSe layers have been deposited using several vacuum- and nonvacuum-based methods. Solar cells built using CZTSe layers fabricated via thermal co-evaporation and direct current (DC) sputtering exhibit power conversion efficiencies (PCE) of 11.6% and 11.95%, respectively [3,4]. Another method to induce CZTSe layer-growth is via the stacking and subsequent selenization of metallic layers. This process has been shown to form materials with a PCE of 9.7% [5]. CZTSe layers grown using solution-based processes have exhibited a PCE of 10.1% [6]. CZTSe thin films developed via the electrochemical route exhibited a PCE of up to 8% [7]. Therefore, CZTSe photovoltaic devices may be capable of attaining high PCE values at a low cost, making CZTSe a promising material for the large-scale production of photovoltaic devices.

Recently, nanoparticle inks prepared using mechanochemical milling have gained considerable attention over other nonvacuum-based processes as a simple, low-cost, and eco-friendly alternative. The nanoparticle-based thin film fabrication process involves the preparation and subsequent deposition of nanoparticle inks, followed by annealing (selenization) in the presence of Se vapor or H_2_Se gas. This procedure is used to prepare a variety of compound materials in numerous applications [8,9,10,11]. Shyju et al. (2015) synthesized CZTS and CZTSe thin films in a single-step, ball milling process and studied their physical properties [12]. Tiwari et al. (2017) studied the thermoelectric properties of CZTSe [9]. Malar et al. (2017) explored the effect of milling conditions on the phase purity of CZTSe thin films using Raman spectroscopy [13]. In 2020, Goyal and Malar obtained single-phase CZTSe and investigated its photo-response [14]. Liu et al. (2020) fabricated an SLG/Mo/CZTSe/CdS/i-ZnO/ZnO:Al(AZO)/Al photovoltaic device with an efficiency of 0.18% [15].

Recently, researchers have demonstrated the effective passivation of the grain boundaries in CZTSe absorber layers with the incorporation of alkali metals, particularly sodium, [16,17,18]. Incorporating alkali metals not only passivates the grain boundaries, surface, and defects but also enhances the crystal quality and carrier concentration of the CZTSe absorber layers [19,20,21,22]. Tampo et al. (2020) investigated the impact of Na incorporation in the CZTSe thin film on the morphological and photovoltaic properties as a function of sodium doping [16]. Kim et al. reported the approach for passivation of grain boundaries and defects in the CZTSe samples with and without Na incorporation [17]. Recently, Rehan et al. systematically examined the effects of sodium doping on the performance of CZTSe solar cells thru different sodium incorporation pathways, such as in-situ sodium doping before and after growing thin films [18].

Herein, we present a simple, cost-effective, and environmentally-friendly technique for the preparation of CZTSe nanoparticles and ink materials using a wet mechanochemical process. Thin films were deposited using formulated ink and sodium solutions via the spin-coating method. The post-annealing effects on the spin-coated CZTSe thin films with and without a sodium solution were characterized using X-ray diffraction (XRD), Raman spectroscopy, field-emission scanning electron microscopy (FE-SEM), energy-dispersive X-ray spectroscopy (EDS), ultraviolet-visible (UV-Vis) spectroscopy, and Hall measurements. The photovoltaic properties of the spin-coated CZTSe thin films were analyzed based on their current density—voltage (J-V) characteristics.

## 2. Materials and Methods

All elemental powders of copper (≥99.5%), zinc (≥99.5%), tin (≥99.5%), and selenium (≥99.99%) as well as chemical reagents, such as 1-butanol (≥99.0%), methyl ethyl ketone (MEK, ≥99.0), Tween-80 (chemically pure grade), and cadmium sulfate (≥99.0%), were purchased from Sigma-Aldrich (Saint Louis, MO, USA). Ethanol (≥99.8%), sodium borohydride (chemically pure grade), and polyethylene glycol (PEG-400, extra pure grade) were purchased from Fluka (Shin-Dong, Korea), Daejung (Siheung-si, Korea) and Samchun Chemicals (Seoul, Korea), respectively. Thiourea (chemically pure grade) and ammonia solution (chemically pure grade) were obtained from Duksan, Seoul, Korea. All chemicals were used as received without further purification. Commercially available Mo-coated soda-lime glass (SLG) (DKG Co., Daegu, Korea) with a thickness of 0.5 mm and resistivity of ~1.0 Ω/sq was used for the thin film preparation.

The as-deposited thin films without and with sodium solution were named Se0 and Se0_Na, while the thin films annealed at 500 °C, 520 °C, and 550 °C without and with sodium solution were named Se1, Se2, Se3 and Se1_Na, Se2_Na, Se3_Na, respectively. As fabricated photovoltaic devices using the above annealed thin films were named SC-Se1, SC-Se2, SC-Se3, SC-Se1_Na, SC-Se2_Na, and SC-Se3_Na.

A detailed description of the preparation of CZTSe nanopowders and ink (Appendix A), and fabrication of corresponding thin films and photovoltaic devices was given in our previous report [23] and Appendix A. Material and device characterization techniques employed in this study were summarized in Appendix A as well.

## 3. Result and Discussion

### 3.1. Zeta Potential and DLS

The dispersion stability profile and hydrodynamic size of the CZTSe ink nanoparticles were determined using zeta potential (dip cell) and dynamic light scattering (DLS) techniques. The results are shown in Figure 1a,b. A mean zeta potential higher than ±30 mV was observed, thereby confirming the stability of the CZTSe nanocrystals in the prepared ink [24,25]. The negative zeta potential suggests the presence of Se^2−^ ions on the surface of the nanoparticles [26]. The average hydrodynamic size of the CZTSe nanocrystals was 913 nm.

### 3.2. Grazing-Incidence X-ray Diffraction (GI-XRD) Analysis

The XRD diffraction patterns of the as-synthesized CZTSe nanocrystals (Se0, Se1, Se2, Se3, Se0-Na, Se1_Na, Se2_Na, and Se3_Na) are depicted in Figure 2a (without thin sodium layers) and Figure 2b (with thin sodium layers). Sharp XRD diffraction peaks are observed at approximately 2θ = 27.19°, 45.25°, and 53.49°, corresponding to d-spacings of approximately 3.276 Å, 2.002 Å, and 1.711 Å, which are attributed to the (112), (204), and (312) planes of the CZTSe kesterite crystal structure (JCPDS card no. 00-052-0868), respectively. Various weak peaks at 2θ = 17.34°, 21.98°, 31.75°, 65.94°, and 72.65°, corresponding to the (101), (110), (008), and (332) planes of the annealed CZTSe samples, respectively, were also observed. In addition to the characteristic peaks of the CZTSe absorber, the primary XRD peak of Mo was detected at 2θ = 40.53° (110) as an internal standard, indicating a well-aligned sample height. All the annealed CZTSe samples exhibited polycrystallinity. No characteristic diffraction peaks of secondary phases were found in any of the prepared CZTSe samples with or without sodium. This confirms the formation of a single-phase for all synthesized CZTSe samples.

Structural properties, such as crystallite size, dislocation density, strain, and lattice constants, of the CZTSe samples, were determined using the XRD diffraction pattern as described in our previous study [23]. The data are summarized in Appendix A. The determined lattice constants (*a* = *b* = ~5.47 Å and *c* = ~11.34 Å) correspond well with the standard lattice constant values of the kesterite CZTSe material (*a* = *b* = 5.69 Å and *c* = 11.33 Å) (JCPDS card no. 00-052-0868). Tetragonal distortion (*c*/2*a* ratio of 1) was observed in all CZTSe samples. These results correspond well with the literature values and further confirm the kesterite structure [14,27]. Using the Debye–Scherrer formula, the crystallite size was estimated to be in the range of 9–45 nm, whereas the average crystallite size obtained using the Williamson–Hall method was approximately 7–50 nm. Smaller crystallite sizes along with higher dislocation densities and strains were observed for the as-synthesized CZTSe as well as Se0 (before annealing). A decrease in the crystallite size and an increase in the strain and dislocation density with increasing temperature were observed, indicating poor crystallinity (Appendix A). The strain of all CZTSe samples showed a negative slope, which suggests lattice compression toward the *c*-axis (Appendix A) [23,28]. The relative intensity ratios of *I*_(112)_/*I*_(204)_ and *I*_(112)_/*I*_(312)_ for Se1 and Se1_Na were observed to be higher than the standard values [*I*_(112)_/*I*_(204)_ = 2.63, and *I*_(112)_/*I*_(312)_ = 5.0] (Appendix A). The relative intensity ratios for the other prepared samples possessed either moderate (Se0, Se2, and Se0_Na) or lower (Se3 and Se3_Na) values compared with those of the standard. A higher intensity ratio indicates that [112] is the preferred orientation for the sample annealed at a lower temperature. This is probably owing to the growth of the MoSe_2_ layer, which mainly affects the texturization of the CZTSe material [29,30].

Notably, the CZTSe XRD diffraction patterns overlap with those of Cu_2_Se, ZnSe, and Cu_2_SnSe_3_. Consequently, the phase purity of CZTSe cannot be confirmed using XRD alone. Raman spectroscopy was, therefore, used to obtain further insights into the phase of the fabricated material.

### 3.3. Raman Analysis

Raman scattering technique was used to confirm the phase purity of all the prepared CZTSe samples, as shown in Figure 3a–e. Peaks were identified at Raman shifts of 187 and 234 cm^−1^ for the as-synthesized CZTSe and Se0 samples, respectively (Figure 3a,b). The intense vibrational bands for the annealed samples (Se1, Se2, and Se3) at different temperatures were observed at ~171, ~193, and ~234 cm^−1^ (Figure 3c–e) [15]. It is currently unknown why two Raman peaks appeared in the samples before annealing, although this has been previously reported [31,32]. The broadening and change in position of the peaks might be due to a change in crystallinity and internal lattice stress [12,33].

The intense vibrational bands at Raman shifts of 171, 193, and 234 cm^−1^ belong to the CZTSe kesterite phase [15]. The strong bands at 171 and 193 cm^−1^ were attributed to the A1 symmetry mode, which correlates to the vibration of chalcogenide (selenium) atoms within the lattice [14]. The peak at 234 cm^−1^ belongs to the B-symmetry mode. These results are comparable to previously reported CZTSe kesterite Raman peaks [13,15,34]. Raman peaks for other secondary phases, such as binary and ternary phases (e.g., ZnSe and Cu_2_SnSe_3_), have been reported and are summarized in Appendix A. The absence of characteristic peaks of the secondary phases in the Raman spectra for the CZTSe confirmed the formation of a single phase. Raman spectroscopy offers key information to enable the successful characterization of the stannite and kesterite structures of the CZTSe materials. The peaks at 209, 226, 232, and 254 cm^−1^ are suggestive of a stannite structure, contrary to the kesterite structure (at 150–200 (171, 187), 216, 230, and 239 cm^−1^). The corresponding peaks at 171, 193, and 234 cm^−1^ of the annealed samples closely resembled the kesterite structure. Despite the peaks at 193 and 234 cm^−1^ being close to the peaks at 209 and 232 cm^−1^ of the stannite structure, no peaks were observed at 254 cm^−1^. These results confirm the formation of single-phase CZTSe kesterite.

### 3.4. FT-IR Analysis

The FT-IR spectra of the CZTSe samples, PEG-400, and Tween-80 in the range of 400–4000 cm^−1^ are shown in Figure 4a,b. Major FT-IR peaks were observed at approximately 3300, 1700, 1600, and 986 cm^−1^ for samples CZTSe, Se0 and Se1. Additionally, a weak absorption band of metal selenide can be observed in the range of 550–400 cm^−1^ for samples CZTSe, Se0 and Se1 [35,36]. The FT-IR peaks of pure PEG-400 [37] and Tween-80 [38] were observed at approximately 3600, 2900, 1452, 1245, 940, and 800 cm^−1^ (Figure 4a). The band at wavenumber ~2900 cm^−1^ corresponds to the C–H stretching vibrations of the –CH_2_ group, and the peak at 1245 cm^−1^ is attributed to C–O stretching vibrations. The peak at 1452 cm^−1^ corresponds to the C–H bending vibrations of the –CH_2_ group. However, asymmetrical bending vibrations of CH_3_ were observed. The peak near 900 cm^−1^ corresponds to C–O–C symmetrical stretching. A characteristic peak at 1735 cm^−1^ corresponding to the C=O group was observed in the FT-IR spectrum of Tween-80. The band at approximately 600 cm^−1^, attributed to the secondary phase (ZnSe), was absent in both Se0 and Se1. The spectra of the Se0 showed absorption bands in regions similar to those of Tween-80 and PEG-400. A possible explanation for this might be that the organic molecules (solvents, binder, and surfactant) used in the ink possess similar FT-IR absorption profiles.

The FT-IR peaks in a similar region were observed for Se2 and Se3 at wave numbers of approximately 3300, 1700, 1600, and 984 cm^−1^, and these observed peaks are well-match with the unannealed samples. A weak vibrational band of metal selenide was observed in the spectrum at 550–400 cm^−1^ in the as-synthesized Se2 and Se3 samples [35,36]. However, characteristic peaks ~600 cm^−1^ for ZnSe were not observed, further confirming the formation of single-phase CZTSe [39]. The peak at ~3400 cm^−1^ is attributed to the chalcogen-rich (selenium-rich) composition. The bands at approximately 1600–900 cm^−1^ belong to the bending and stretching vibrational frequencies of oxygen. These bands arise from hygroscopic materials and surface-adsorbed moisture; that is, H_2_O and CO_2_ molecules, contributing O-H and C-O-associated peaks in the FT-IR spectrum [39]. FT-IR analysis also confirms the presence of a pure phase in all prepared CZTSe samples.

### 3.5. FE-SEM and EDS Analyses

The effects of annealing and sodium incorporation on the surface morphology of the CZTSe samples were investigated using FE-SEM. The morphology and cross-sectional images along with the EDS data of Se0, Se1, Se2, and Se3 without a sodium layer are presented in Figure 5, Figure 6, Appendix A, respectively. Appendix A show FE-SEM images of Se0 and Se0_Na thin films, respectively. The surface images show that the Se0 and Se0_Na thin films consisted of nanoparticles and organic residues (PEG-400 and Tween-80), suggesting that the organic additives used for the effective spin coating of the ink were not completely eliminated during the pre-heating step. The minimum temperature for decomposing these molecules into vaporized phases has been reported to be approximately 250 °C [40,41]. FE-SEM images of the Se0 and Se0_Na thin films showed several voids with small grain sizes on the surfaces. FE-SEM images of the Se1, Se2, and Se3 thin films are presented in Figure 5, Figure 6 and Appendix A, respectively. During annealing, the organic molecules can be thermally broken down into smaller components and vaporized at high temperatures (500–550 °C). However, some voids with discontinuous grain growth were observed on the surfaces of the samples prepared using this annealing process. Similar surface properties of thin films prepared using nanoparticle ink have also been observed in previous studies [42].

The impact of sodium incorporation on the morphology of the Se1_Na, Se2_Na, and Se3_Na thin films was investigated using FE-SEM, as shown in Figure 7, Figure 8 and Appendix A. The surface images show that Se1_Na, Se2_Na, and Se3_Na thin films were non-homogenous, with pinholes and voids, and demonstrated varying grain sizes with discontinuous growth. Although the surface morphology of the thin films is uneven and non-compact, it can aid in evaporating organic materials and reducing carbon residues from the bottom of thin films [43]. The thicknesses of the CZTSe thin films with and without the sodium layer as well as before and after annealing was observed to be in the range of 900 nm to 2.0 μm. Significantly, surface imperfections, such as cracks, voids, and holes, in thin films can increase the series resistance, leading to recombination sites and/or leakage paths in photovoltaic devices.

The formation of a MoSe_2_ layer near the interface of the absorber and rare contact layers was observed in the cross-sectional images of all annealed CZTSe samples. Similar to the surface properties of the CZTSe, voids, cracks, and holes were detected on the surface of the thin films during grain growth at different annealing temperatures. These voids, cracks, and holes likely provided channels for the selenium vapors to diffuse. The MoSe_2_ layers in Se1, Se2, and Se3 thin films had a thickness of 95, 109, and 178 nm, respectively. In the case of the Se1_Na, Se2_Na, and Se3_Na thin films, the MoSe_2_ layers had a thickness of 278, 317, and 377 nm, respectively. The thickness of the MoSe_2_ layer was observed to increase with increasing annealing temperature for all CZTSe samples. This may have resulted from the enhanced Se diffusion owing to the poor surface morphology.

SEM-EDS was used to investigate the chemical composition of the CZTSe; the findings are summarized in Table 1, Appendix A. The chemical composition profiles of all CZTSe samples were slightly different from each other. A trend with an increasing ratio of Se/(Cu + Zn + Sn) in the CZTSe samples was observed, indicating the substantial incorporation of selenium into the thin films prepared without sodium. The CZTSe thin films prepared with sodium showed an initial decrease, followed by an increase in the Se/(Cu + Zn + Sn) ratio with increasing temperature. The ratios of Cu/(Zn + Sn) and Zn/Sn for all CZTSe samples showed a moderate deviation from the stoichiometric value.

### 3.6. Bulk XRD of Cu_2_ZnSnSe_4_ Thin Films

To confirm the formation of MoSe_2_ at the interface of CZTSe and Mo in the prepared thin films, bulk XRD measurements were conducted. As shown in Figure 9a,b, in addition to the characteristic peaks of the CZTSe thin films, sharp peaks corresponding to Mo were identified at 2θ ≈ 40.56° (110) and 73.62° (211), and slightly broad MoSe_2_ peaks were observed at 2θ ≈ 31.79° (100) and 56.36° (110). [44,45]. The relative intensity of the XRD peaks of MoSe_2_ increased with increasing annealing temperature, presumably because of the enhanced selenization of Mo at high temperatures.

### 3.7. Optical Properties

The bandgap of the CZTSe samples was determined by plotting *(αhv)^x^* as a function of incident photon energy (*hv*, eV) (Equation (1)). The linear portion of the spectrum in the high-absorption regime was then extrapolated, yielding the bandgap at the intercept with the photon energy axis [46]:(1)$$\alpha hv=A{\left(hv-E_{g}\right)}^{x}$$
Where *α*, *h*, *v,* and *E_g_* are the absorption coefficient, Planck’s constant, light frequency, and optical bandgap (eV), respectively. *A* and *x* are both constants. The *x* value depends on the direct or indirect transition of semiconductor materials. A UV-Vis-NIR spectrophotometer was used to obtain the transmission spectra of the thin films. The absorption coefficient was evaluated using Equation (2) [42]:(2)$$\alpha =\frac{1}{d}\ln\left(\frac{1}{T}\right)$$
where *α* is the absorption coefficient, and *d* and *T* are the thickness and transmittance of the CZTSe thin film, respectively.

The optical properties of the prepared CZTSe samples were evaluated using transmission spectra in the range of 300–2500 nm, as shown in Figure 10a. The optical properties of semiconductor materials are strongly influenced by surface morphology, explaining the differences in optical properties of CZTSe absorber samples before and after annealing at various temperatures [46]. The optical bandgap values were estimated to be in the range of 0.82–1.02 eV (Figure 10b), which corresponds with the values reported in previous publications [14,15,27,47].

### 3.8. Electrical Properties

The Hall effect using the Van der Pauw method (with a magnetic field intensity of 0.55 T under ambient conditions) was used to investigate the electrical properties. To investigate the impact of annealing and sodium incorporation on the electrical properties, CZTSe thin films were prepared as previously reported [23]. The results are summarized in Table 2.

The carrier concentration and mobility of CZTSe samples were measured to be 6.14 × 10^17^–1.61 × 10^19^ cm^−3^ and 1.13–6.71 cm^2^/V·s, respectively. These values are similar to published reports (10^15^ cm^−3^–10^19^ cm^−3^ and 0.1–10 cm^2^/V·s, respectively) [12,48,49,50]. Relatively high resistivities (0.1 to 5.8 Ω·cm) were measured, possibly resulting from the uneven grain growth and the porous nature of the thin films. Se3_Na exhibited the highest carrier concentration and lowest resistivity with moderate mobility, although void formation and isolated grain growth were observed in the thin film. All CZTSe samples showed p-type conductivity. Although slight variations in the electrical properties of the CZTSe samples were observed when compared with the reported values of mobility (0.1–10 cm^2^/V·s), carrier density (10^15^–10^19^ cm^−3^), and resistivity (10^−2^–10 Ω·cm) [12,27,51,52], however, improvement can be expected by increasing the quality of the films (Table 2).

The results from various analytical techniques, including GI-XRD, Raman, FT-IR spectroscopy, FE-SEM-EDS, UV-Vis-NIR spectrophotometry, and the Van der Pauw measurements, suggest that CZTSe thin films prepared using the nanoparticle ink method can potentially be used as absorber layers to fabricate photovoltaic devices.

### 3.9. Photovoltaic Analysis: J-V Characteristic

The CZTSe absorber layers with and without sodium were integrated into the photovoltaic device in the traditional configuration of SLG/Mo/CZTSe:Na or CZTSe/CdS/i-ZnO/AZO/Ni-Ag. The photovoltaic performance for all CZTSe photovoltaic devices, where the total active area of each device was 0.40 cm^2^, (measured under 1 sun, AM1.5 irradiation conditions) is presented in Figure 11a and Table 3. Se2 and Se1_Na exhibited the best results among all devices. Se2 had the maximum efficiency (ƞ = 0.16%) and short-circuit current density (J_sc_ = 1.62 mA/cm^2^), while Se1_Na showed the highest open current-voltage (V_oc_ = 335 mV) and fill factor (FF = 44.19%). However, the photovoltaic performance of the CZTSe devices (with and without sodium) fabricated in this study was significantly lower compared with that of pure CZTSe devices fabricated using the vacuum-based technique [3,53] and toxic hydrazine-based method [6,54]. The possible reasons for low performance in this study may be as below: (i) the presence of small grains with discontinuous growth and rough surface morphology of the CZTSe samples, (ii) defects, (iii) mismatched energy band alignment at interfaces, (iv) poor crystallinity as increasing the annealing temperature, (v) low or high carrier density, and (vi) formation of MoSe_2_ layer at the interface of CZTSe and Mo layers. The high series resistance of thick MoSe_2_ can restrict the ability of the Mo electrode to collect photogenerated hole carriers [15,55,56,57]. This could result in the formation of defective recombination centers, thereby hindering the transport and collection of photogenerated charge carriers and ultimately causing low-performance parameters in the fabricated photovoltaic devices. These findings are being subjected to further investigation in our laboratory to improve the performance of CZTSe-based photovoltaic devices. Figure 11b,c shows cross-sectional images of the photovoltaic devices of Se2 and Se1_Na, respectively.

The following strategies have been used in the literature for improving the efficiency of CZTSe cells: (i) Surface treatments or surface passivation: after sintering, the appearance of a high conducting phase, such as Cu_x_Se_y_ on the surface of the CZTSe NPs absorber layer shows a detrimental impact. Therefore, surface treatment, such as KCN etching is necessary to remove those secondary phases. Further, the passivation of CZTSe is necessary. (ii) In order to promote the crystallization, a high-temperature annealing step can be employed, and hence annealing under sulfurization or selenization is essential to increase the grain growth. (iii) The formation of MoSe_2_ can be reduced by passivation at the back contact interface. (iv) The deposition and pre-heating conditions can be optimized for the evaporation of the solvent and/or additives since too high drying temperature and speed can cause initial cracks in the deposited films. (v) Carrier doping density can be further optimized. (vi) A Cu-deficient (Cu/(Zn + Sn)~0.8) and Zn-rich (Zn/Sn~1.2) compositions are important for highly efficient photovoltaic devices, and therefore, the elemental composition of CZTSe samples requires further optimization [42,57,58].

## 4. Conclusions

In this study, single-phase CZTSe thin films were deposited using a simple, reproducible, and cost-effective colloidal ink method, that is, a non-vacuum synthesis technique. The kesterite CZTSe nanoparticles and ink were prepared using a mechanochemical milling process. CZTSe thin films were deposited from the prepared nanoparticle inks via spin-coating using inexpensive and innocuous azeotropic solvents (ethanol and MEK) under ambient conditions. The influence of different annealing temperatures (500–550 °C) and sodium incorporation on the photovoltaic performance and material properties, such as surface morphology, chemical composition, phase purity, optical properties, and electrical properties, were evaluated. The outcomes of material analyses support the successful formation of CZTSe in the single phase. The CZTSe thin films possessed small porous grains with discontinuous growth, regardless of the annealing temperature. EDS analysis showed the substantial incorporation of chalcogen (Se) into the absorber films during annealing at different temperatures. All the obtained CZTSe samples showed p-type conductivity with an optical bandgap between 0.82–1.02 eV. The efficiencies of the photovoltaic device fabricated from Se2 and Se1_Na were 0.16% and 0.12%, respectively. This simple and effective fabrication approach provides a low-cost and controlled method to produce CZTSe samples without the use of hazardous and toxic gases, dangerous solvents, or complicated vacuum equipment.

## Figures and Tables

**Figure 1 nanomaterials-12-01503-f001:**
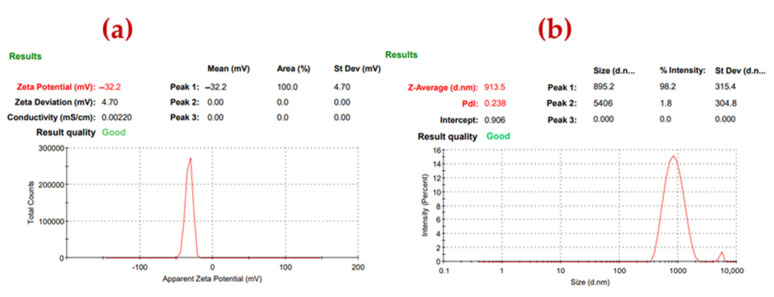
(**a**) Zeta potential and (**b**) DLS of CZTSe ink.

**Figure 2 nanomaterials-12-01503-f002:**
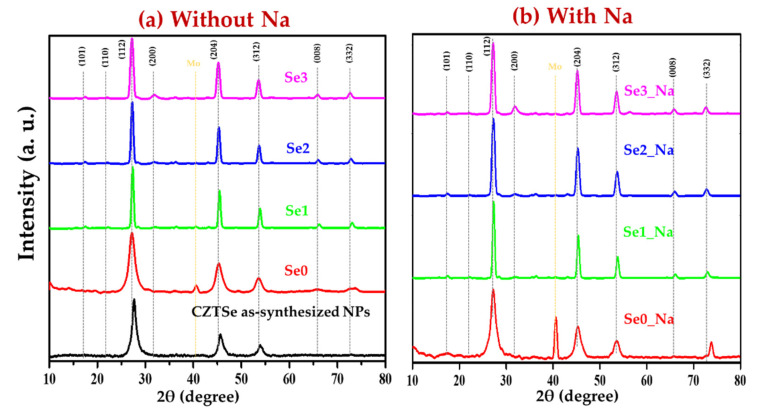
XRD pattern of CZTSe thin films (**a**) without and (**b**) with sodium layer.

**Figure 3 nanomaterials-12-01503-f003:**
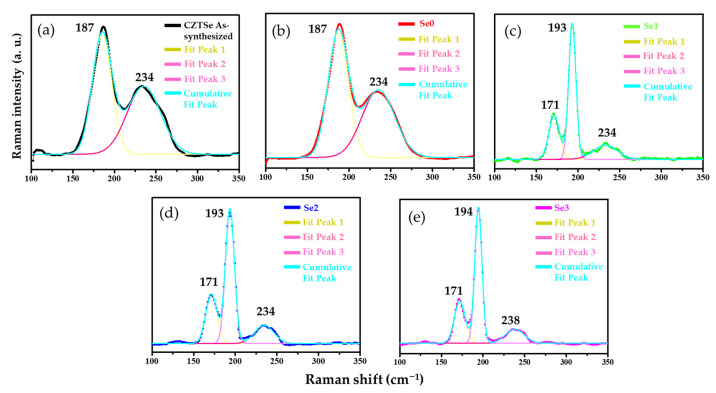
Raman Spectra of CZTSe thin films: (**a**) As-synthesized CZTSe NP, (**b**) as-deposited CZTSe thin film (Se0), (**c**) Se1 (annealed at 500 °C), (**d**) Se2 (annealed at 520 °C), and (**e**) Se3 (annealed at 550 °C).

**Figure 4 nanomaterials-12-01503-f004:**
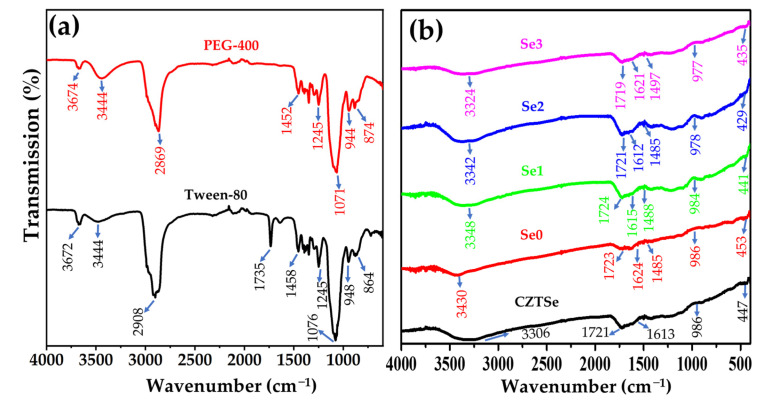
FT-IR spectra of (**a**) Tween-80 and PEG-400, and (**b**) CZTSe thin films.

**Figure 5 nanomaterials-12-01503-f005:**
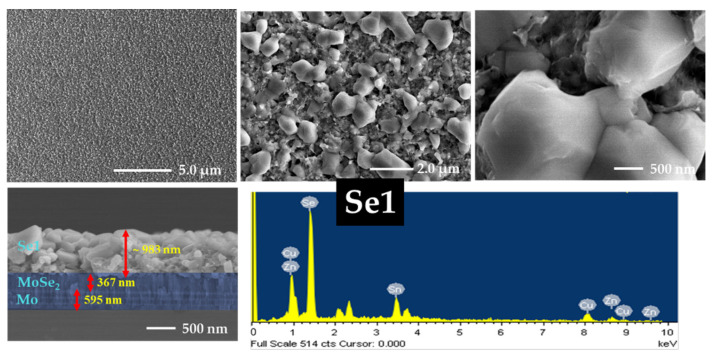
Surface and cross-section images and EDS spectra of Se1 thin film.

**Figure 6 nanomaterials-12-01503-f006:**
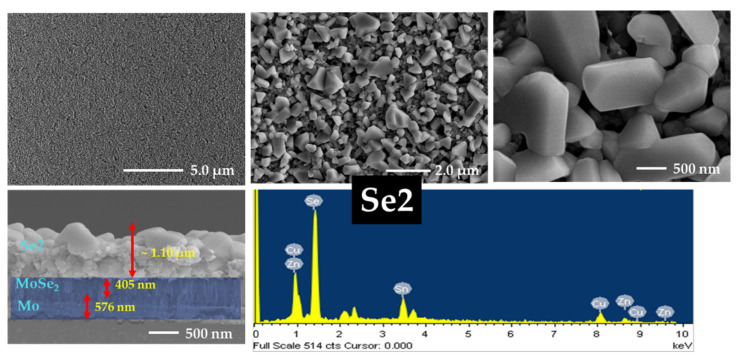
Surface and cross-section images and EDS spectra of Se2 thin film.

**Figure 7 nanomaterials-12-01503-f007:**
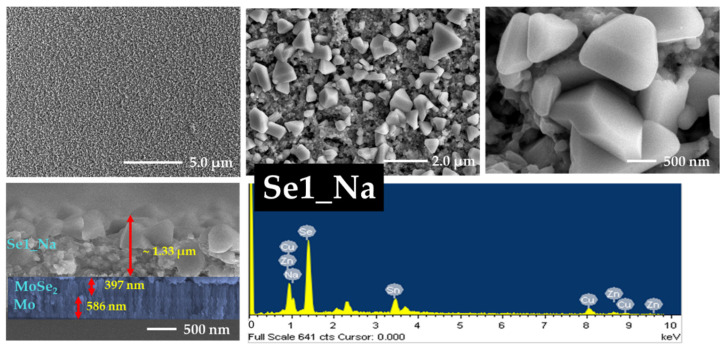
Surface and cross-section images and EDS spectra of Se1_Na thin film with sodium layer.

**Figure 8 nanomaterials-12-01503-f008:**
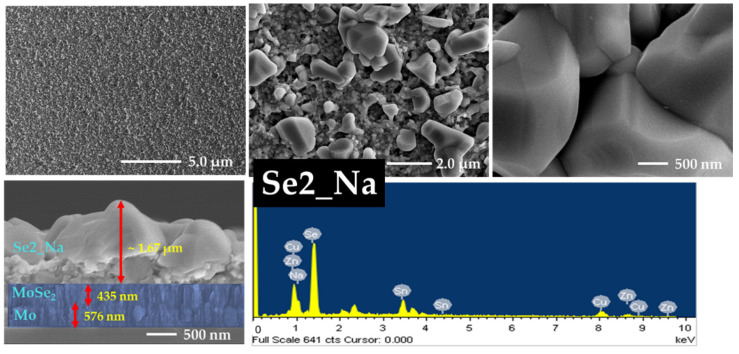
Surface and cross-section images and EDS spectra of Se2_Na thin film with sodium layer.

**Figure 9 nanomaterials-12-01503-f009:**
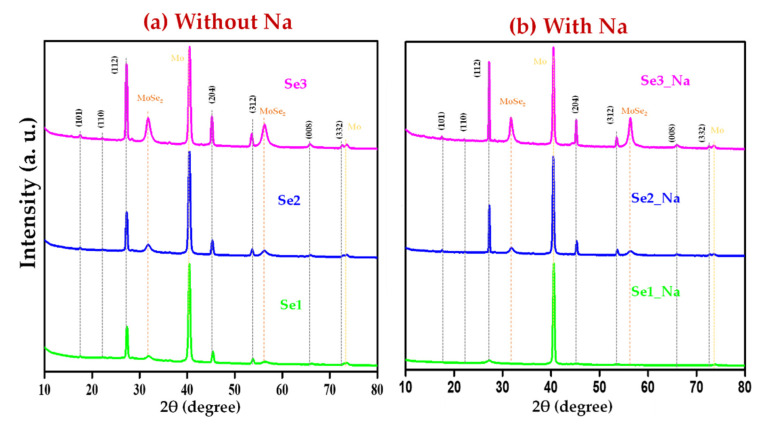
Bulk XRD patterns of CZTSe thin films (**a**) without and (**b**) with sodium layer.

**Figure 10 nanomaterials-12-01503-f010:**
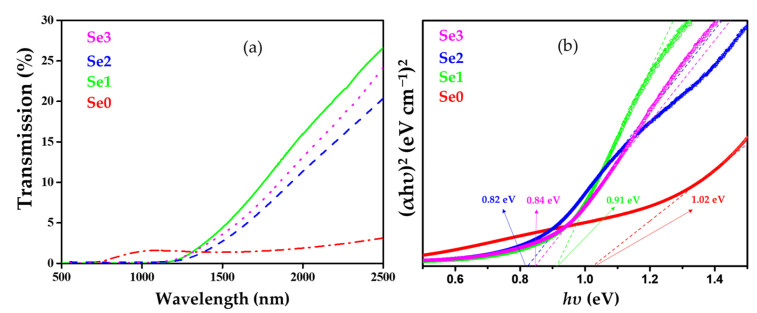
(**a**) Transmission spectra and (**b**) bandgap of CZTSe thin films.

**Figure 11 nanomaterials-12-01503-f011:**
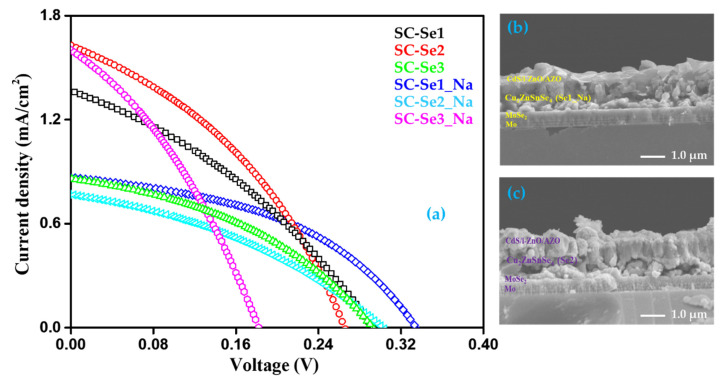
(**a**) J-V curve of CZTSe photovoltaic device under AM 1.5G illumination, and cross-section image of (**b**) Se2 and (**c**) Se1_Na photovoltaic devices.

**Table 1 nanomaterials-12-01503-t001:** Elemental composition of CZTSe without and with sodium layer.

Sl No.	Sample Name	Cu/Zn + Sn	Zn/Sn	Se/(Cu + Zn + Sn)
1	Se0	1.05	1.07	0.86
2	Se1	0.95	1.19	0.85
3	Se2	1.07	1.02	0.89
4	Se3	0.91	1.32	0.92
5	Se0_Na	1.12	0.82	1.03
6	Se1_Na	1.22	0.83	0.89
7	Se2_Na	1.17	0.78	0.92
8	Se3_Na	1.12	1.05	0.98

**Table 2 nanomaterials-12-01503-t002:** Electrical properties of CZTS thin films with and without Na layer.

Sl. No.	Sample Name	Carrier Conc. (cm^−3^)	Mobility cm^2^/V·s	Resistivity (Ω cm)
1	Se0	2.10 × 10^18^	1.21	2.47
2	Se1	1.07 × 10^18^	1.33	4.41
3	Se2	1.05 × 10^18^	1.54	3.87
4	Se3	1.80 × 10^18^	1.13	3.06
5	Se0_Na	1.13 × 10^18^	1.12	4.96
6	Se1_Na	2.04 × 10^17^	6.71	4.56
7	Se2_Na	6.14 × 10^17^	1.76	5.77
8	Se3_Na	1.61 × 10^19^	3.86	0.101

**Table 3 nanomaterials-12-01503-t003:** Photovoltaic properties of CZTSe thin films with and without sodium layer.

Sl No.	Sample Name	V_oc_ (mV)	J_sc_ (mA/cm^2^)	FF (%)	η (%)
1	SC-Se1	292	1.37	36.56	0.14
2	SC-Se2	266	1.62	37.79	0.16
3	SC-Se3	293	0.85	41.43	0.10
4	SC-Se1_Na	335	0.86	44.19	0.12
5	SC-Se2_Na	300	0.76	41.32	0.10
6	SC-Se3_Na	182	1.60	33.55	0.09

## Data Availability

Not applicable.

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
