# Peer review of "Synthesis and Characterization of Cu_2_ZnSnSe_4_ by Non-Vacuum Method for Photovoltaic Applications"

_nanomaterials, 2022, doi:10.3390/nano12091503_

Round 1
Reviewer 1 Report
The manuscript is devoted to the formation and characterization of Cu2ZnSnSe4 by the non-vacuum method expecting that such material would be suitable for photovoltaic applications. The film formation includes ball mailing, annealing in the proper atmosphere and spin coating. The method is declared a “simple and effective fabrication approach that provides a low-cost and controlled method to produce CZTSe samples without the use of hazardous and toxic gases, dangerous solvents, or complicated vacuum equipment”. However, the obtained thin films are porous, with uneven grains, consisting of several components. Using these films, even with the addition of some other elements (Na), yields in solar cells with more than modest efficiency (0,14-0,16%) compared with the same device with material produced by other techniques (8-10%). Reading the actual manuscript, it is not clear if such results are due to the method of preparation, or the chosen parameters were not good enough.
The manuscript is well organized, and the experiment is described in detail. Discussion is correct and convincing while limited to explaining only the results of measurements.
Final version of manuscript should contain discussion/explanation why solar cells with obtained material exhibits such low efficiency and in which way is possible to improve its properties using the proposed method of preparation.
Some additional remarks
292 “sharp peaks corresponding to Mo” should be MnSe2
318 “the variation in the optical properties of the absorber samples before and after annealing at various temperatures” English
There are no units on the graph in Fig 10b
Something Is wrong with table S2 in Supplement (4 columns but 1 title)
Conclusions
“Data from XRD, Raman, and FT-IR analyses indicates the successful formation of CZTSe in the pure phase.”
How the samples are “pure phase” and in the same time they consist of several components?
Reviewer 2 Report
The Cu2ZnSnSe4 has been synthesized and characterized by non-vacuum method for photovoltaic applications. This method is simple and low-cost. However, the photovoltaic devices exhibit not very good current-voltage characteristics. I suggest that this manuscript can be accepted with minor revised. The issues are as followed.
- Introduction section, how about the references of adding the alkali metals in the latest two year?
- Since the photovoltaic performance of the CZTSe devices (with and without sodium) fabricated in this study was significantly lower compared with that of pure CZTSe devices fabricated using the vacuum based technique, I wonder how to improve the performance of this ink sample with sodium?
- Why the XRD pattern of Se1-Na is different from the other two samples(Se2-Na, Se3-Na)?
Reviewer 3 Report
- Commercially available Mo-coated soda-lime glass (SLG) with a thickness of 0.5 mm. What was the resistivity of this?
- Se0, Se1, Se2, Se3, should be defined at the start of the manuscript, not in the supplement. file.
- What annealing temperatures were used should provide in the main manuscript, not in the supplement file.
- What is the reason for peak shifting in FTIR at lower wavelengths.?
- What is the effect of Na doping on FTIR and Raman properties?
- In table2, the electrical properties of Se0 should be added.
- What is the novelty of this research as the current work shows lower performance as compared to reported data and most of the experimental work is carried out or cited by the group's previous work?
Round 2
Reviewer 1 Report
I have no further remarks.
Reviewer 3 Report
The authors have revised the manuscript as per suggestions.